# λGrapher: A Resource-Efficient Serverless System for GNN Serving through Graph Sharing

## ABSTRACT

Graph Neural Networks (GNNs) have been increasingly adopted for graph analysis in web applications such as social networks. Yet, efficient GNN serving remains a critical challenge due to high workload fluctuations and intricate GNN operations. Serverless computing, thanks to its flexibility and agility, offers on-demand serving of GNN inference requests. Alas, the request-centric serverless model is still too coarse-grained to avoid resource waste.

Observing the significant data locality in computation graphs of requests, we propose λGrapher, a serverless system for GNN serving that achieves resource efficiency through graph sharing and fine-grained resource allocation. λGrapher features the following designs: (1) adaptive timeout for request buffering to balance resource efficiency and inference latency, (2) graph-centric scheduling to minimize computation and memory redundancy, and (3) resource-centric function management with fine-grained resource allocation catered to the resource sensitivities of GNN operations and function orchestration optimized to hide communication latency. We implement a prototype of λGrapher based on the representative open-source serverless platform Knative and evaluate it with real-world traces from various web applications. Our results show that λGrapher can achieve savings of up to 54.2% in memory resource and 45.3% in computing resource compared with the state-of-the-art while ensuring GNN inference latency.

## 1 INTRODUCTION

Graphs, as a fundamental data structure, are prevalent in various domains including social networks [29, 46], financial networks [5, 36], and transportation networks [15, 30]. The rise of deep learning has empowered graph neural networks (GNNs) to be a powerful tool to extract features from graph structures [40, 41, 50]. Today, GNNs have been widely used in online web services, e.g., social network analysis [9, 23], short-video recommendation [25, 47], shopping recommendation [35, 44], and financial fraud detection [26, 36].

However, efficient serving of GNNs—running GNNs for time-sensitive inference tasks—remains a critical challenge, for the following reasons: (1) GNNs are computation- and memory-intensive due to the large graph size and complex operations, while applications impose stringent service-level objectives (SLOs) on GNN inference latency [48]. (2) The arrival of GNN inference requests in web services is typically busty and hard to predict [45]. (3) GNN execution intricately interleaves graph and tensor operations that show diverging resource sensitivities [33]. The resource inefficiency of GNN deployment leads to high operational costs for web services.

To deal with workload fluctuations, web services typically adopt autoscaling techniques to adjust the provisioned resources vertically and horizontally. Specifically, the system monitors a metric such as the CPU or memory utilization and applies a threshold-based scaling policy [4, 13]. Upon workload increases and the utilization exceeds the threshold, a more powerful service instance (e.g., with more CPU cores or memory) is launched to replace the current one in the case of vertical scaling, or more service instances are added to serve requests in the case of horizontal scaling. The opposite will be applied when the workload decreases and the utilization drops below the threshold. While such autoscaling techniques can absorb workload variations at large time scales, the long delay in changing the provisioned resources (e.g., launching new virtual machines) limits their capability of handling short-term request spikes.

Serverless computing (and its popular implementation function as a service) offers new opportunities for efficient provisioning of web services thanks to its agile event-driven model [16]. However, a direct request-centric serverless deployment of GNN inference, i.e., invoking a separate function to process each arriving request, as done in financial fraud detection systems on AWS Lambda [6], may not provide us with the promised efficiency gain. There are two major reasons: (1) The fixed resource allocation for a function invocation per request ignores the diverging resource sensitivities of operations in different GNN execution stages, leading to low overall resource utilization. (2) Per-request function innovation leads to repeated computation and redundant memory usage across requests that potentially share parts of their computation graphs.

In this paper, we present λGrapher, a scalable, resource-efficient serverless system for GNN inference. Our key observation is that GNN inference requests arriving in a given period show high spatial data locality, i.e., their computation graphs overlap significantly. Following this observation, λGrapher features the following designs to achieve high resource efficiency: First, λGrapher buffers requests and processes them in batches to exploit the data locality and reduce computation and memory redundancy. As request buffering introduces extra delay, to strike a good balance between resource efficiency and latency, λGrapher incorporates *adaptive timeout configuration* to decide when the batch of requests in a buffer must be dispatched to avoid latency SLO violation. Second, λGrapher adopts *graph-centric scheduling* to perform GNN inference computation. Specifically, we use multiple queues and distribute arriving requests to these queues, aiming to maximize the spatial data locality of requests in the same queue. To execute the aggregate computation of batched requests, we merge the computation graphs of all these requests and partition the merged graph accounting for locality so that resources allocated for a partition can be released immediately once the local computation is completed, leveraging the agility of serverless functions. Finally, λGrapher employs *resource-centric function management* which allocates resources to functions catering to the resource sensitivities of the GNN operations performed by each function and orchestrates functions into a pipeline to reduce inter-function communication time overhead.

In short, this paper makes the following contributions. After conducting a thorough empirical analysis of GNN workload variations, data locality, and resource sensitivities of GNN operations (§2), we

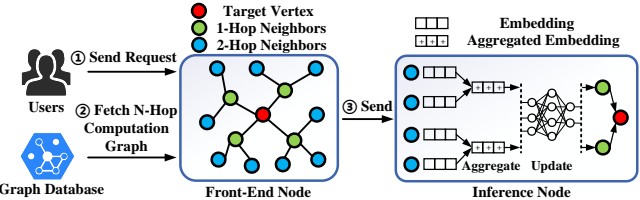

Figure 1: Typical GNN inference workflow.

- present a resource-efficient serverless system for GNN inference (§3) featuring an adaptive timeout mechanism for request buffering to balance resource efficiency and end-to-end latency.
- propose a graph-centric request scheduler that exploits data locality to minimize computation and memory redundancy and maximize resource elasticity.
- introduce a resource-centric function manager that caters the resource allocation to the specific resource sensitivities of GNN operations and orchestrates functions in pipelines to reduce inter-function communication latency.
- implement $\lambda$Grapher on the serverless platform Knative. Our evaluation with real-world request traces shows that $\lambda$Grapher achieves savings of up to 54.2% in memory resource and 45.3% in computing resource when compared to the state-of-the-art (§4). §5 discusses related work and §6 concludes the paper.

## 2 BACKGROUND AND MOTIVATION

This section describes the fundamentals of GNN and the inference workflow in current systems, empirically studies the workload fluctuations of GNN inference, motivates a graph-centric serverless approach for GNN inference, and discusses the challenges in building an efficient graph-centric serverless system for GNN inference.

### 2.1 Fundamentals of GNN Inference

**GNN basics.** Denote the input graph as $G = (V, E)$, where $V$ is the set of vertices representing specific entities and $E$ is the set of edges representing relationships between entities. Each vertex $v \in V$ has a feature representation $h_v \in \mathbb{R}^d$, where $d$ is the feature dimension. A GNN contains multiple layers, each comprising Aggregate and Update operations. In each layer, every vertex $v$ aggregates information from its neighboring vertices with

$$h_v^{l+1} = \Phi^l\left(\{h_u^l : u \in \mathcal{N}(v)\}\right), \qquad (1)$$

where $h_v^{l+1}$ is the representation of vertex $v$ in layer $l + 1$, $\mathcal{N}(v)$ is the set of neighbors of vertex $v$, $h_u^l$ is the representation of neighboring vertex $u$ in layer $l$, and $\Phi^l$ is the aggregation function. The representation of each vertex $v$ is updated after each layer $l$ with

$$h_v^{l+1} = \Upsilon^l(h_v^{l+1}, h_v^l), \qquad (2)$$

where the update function $\Upsilon^l$ typically includes neural network layers used to integrate information from the current layer and the previous layer, resulting in a new representation for the vertex.

**GNN inference workflow.** GNN inference has been employed by various time-sensitive online services, such as GraphLearn [1] and PlatoGL [25]. Figure 1 shows a typical GNN inference workflow. First, the request content is extracted as vertices and edges. Next, the platform sets the vertex to predict as the target vertex and

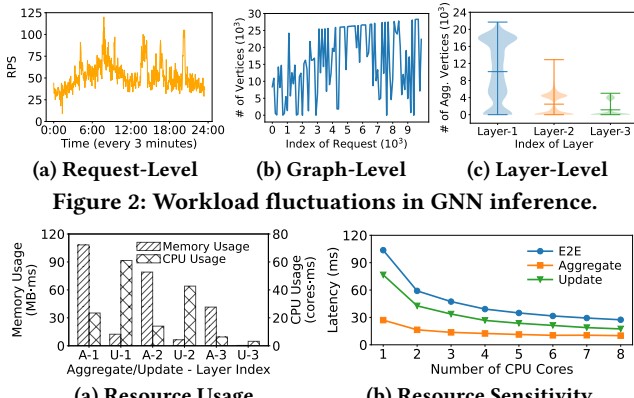

Figure 2: Workload fluctuations in GNN inference.

(a) Resource Usage

(b) Resource Sensitivity

Figure 3: Varying resource sensitivity of GNN operations. The experiment is conducted with a 3-layer GCN.

extracts an $n$-hop computation graph. Then, the feature vectors are extracted following the vertices/edges in this graph. Finally, this graph and feature vectors are used as inputs for inference.

### 2.2 Resource Inefficiency in Current Systems

Current GNN inference systems fall into two types: traditional elastic cloud systems and request-centric serverless systems. The former has pre-configured resources and applies autoscaling in coarse grains based on monitoring metrics, as explained before. Examples of this type include Alibaba's GraphLearn [1] and Tencent's PlatoGL [25]. Request-centric serverless systems handle each request by triggering a function invocation, allowing on-demand processing based on the specific computation graph of the request. AWS's financial fraud detection system operates in this way [6]. Unfortunately, both types of systems suffer from resource inefficiency for one or both of the following two reasons.

**Multi-scale workload fluctuations in GNN inference.** Using widely recognized datasets of user request arrival traces from Twitter [2] and datasets of requests on social network graphs from Twitter [7], we show that the GNN inference workload fluctuates at three levels: request, graph, and layer. Request-level fluctuations are represented by burstiness in the user request intensity, measured by requests per second (RPS), as shown in Figure 2a. Graph-level fluctuations concern the size of the extracted computation graph of each request. We use a typical setup of a 3-hop computation graph from the target vertex for real-time inference and compare the graph size difference between any two consecutively arriving requests. Figure 2b shows the difference can be as large as 98.6×. Layer-level fluctuations are represented by the difference in the number of vertices at each GNN layer, demanding varying resources to perform computation. Figure 2c shows that this difference can reach 4× between Layers 1 and 2 and 9× between Layers 1 and 3.

**Varying resource sensitivity of GNN operations.** Each GNN layer is composed of two main operations alternatively executed: Aggregate and Update. Figure 13 (in Appendix A) shows the the structure of three classic GNN layers, namely GCN [18], GraphSAGE [11], and GIN [42]. Taking a 3-layer GCN model as an example, we investigate the demands and sensitivities of Aggregate and Update to different resource types. Figure 3a shows that Aggregate, a graph-based operation, is memory-bound, whereas Update,

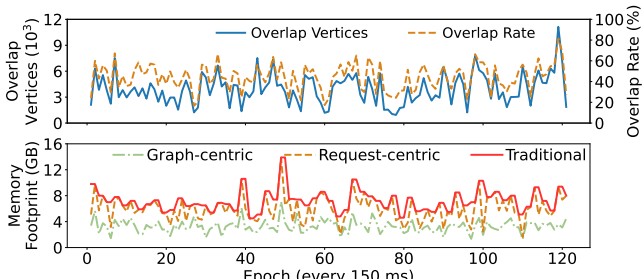

Figure 4: Computation graph overlap among requests over a period and comparative resource consumption analysis of traditional, request-centric, and graph-centric approaches.

a tensor-based operation, is CPU-bound. Figure 3b shows that with the increase of CPU cores, Aggregate shows a continuous latency reduction (up to 4.5×) while the latency for Update quickly plateaus with a maximum reduction of 2.2×. This implies that Aggregate is more sensitive to CPU resource than Update.

Existing systems consider request-level workload fluctuations at best and none of them consider multi-scale workload fluctuations and varying resource sensitivities of GNN operations.

## 2.3 New Opportunities

The above analysis motivates us to switch from the request-centric serverless design to a *graph-centric* one. This design choice offers the following new opportunities.

**Exploiting data locality for graph sharing.** Based on the Twitter trace [2, 7], we observe a significant overlap between the computation graphs of requests arriving within a period. Figure 4 shows that the overlap rate can reach 44.2% for epochs of 150 ms, leading to considerable redundant computation and memory usage, which can be avoided by batching requests and sharing intermediate results across requests [41]. We show in Figure 4 that a graph-centric serverless approach could save, on average, 55.3% and 46.5% memory resource compared with the traditional and request-centric serverless approaches, respectively. Figure 5 shows the resource consumption of two consecutive requests under different execution modes. It shows that batching requests and eliminating redundancy reduces 21.3% of memory usage and 22.7% of CPU usage.

**Decoupling GNN operations for fine-grained resource allocation.** The sensitivity of Aggregate and Update to resources differs, suggesting a resource-centric approach to function management. Specifically, we can manage functions in resource groups, decoupling memory-sensitive Aggregate and compute-sensitive Update and customizing fine-grained resource allocation for each of them. Figure 6 shows that with this approach up to 52% memory reduction and 25% CPU reduction can be achieved (see the "3+3" mode). On the other hand, we pay the cost of slight latency increases, primarily caused by the inter-function communication overhead.

## 2.4 Design Challenges

The graph-centric serverless approach with fine-grained resource allocation offers tremendous benefits, but also raises challenges.

**C1: How to batch requests to exploit data locality?** Request batching is a de-facto optimization in inference serving systems for improving resource efficiency. However, due to the heterogeneity

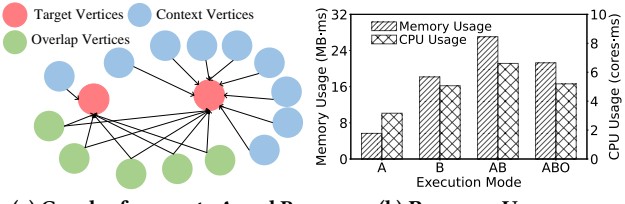

(a) Graph of requests A and B  (b) Resource Usage

Figure 5: Resource utilization under different execution modes. "A" and "B" represent individual executions, "AB" represents batch processing without sharing, and "ABO" represents batch processing with sharing exploiting the overlap.

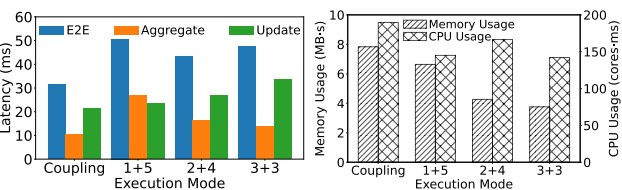

(a) Latency  (b) Resource Usage

Figure 6: Latency and resource comparison between decoupled and coupled groups. Decoupled group "i+j" means allocating i CPU cores to Aggregate and j CPU cores to Update.

of request graphs and irregular memory access in the Aggregate operation in GNN inference (see Figure 2b), batch processing can be inefficient if not treated carefully. As we have shown significant resource efficiency improvement can be achieved by reusing intermediate results among batched requests. The challenge is on quickly grouping requests to maximize the chance of reuse.

**C2: How to efficiently execute batched requests?** When batching requests, the computation graphs of these requests are merged into a big graph, e.g., with millions of vertices. The memory needed to host the merged graph can easily exceed the memory limit of serverless functions, leading to scalability concerns. A quick idea is to break down the merged graph into pieces and allocate a function for each piece. The challenge is on partitioning the merged graph at a suitable granularity to ensure scalability and take advantage of the agility of serverless functions to achieve resource efficiency.

**C3: How to conceal inter-function communication overhead?** Decoupling GNN operations and enabling fine-grained resource allocation offers efficiency gains, but at the cost of extra inter-function communication overhead. One typical approach is to construct a pipeline to overlap function execution with communication. The challenge is to fine-tune this pipeline so that all the functions in the pipeline achieve load balancing to maximize overhead hiding.

## 3 SYSTEM DESIGN

We present λGrapher and its design in this section.

## 3.1 System Overview

To address the shortcomings of the request-centric serverless service model discussed in Section 2.2, we develop a resource-efficient serverless GNN inference system with a graph scheduling and resource management engine. The main idea behind the engine lies in two aspects: (1) graph-centric scheduling which leverages the graph sharing of consecutively arriving requests to reduce computation

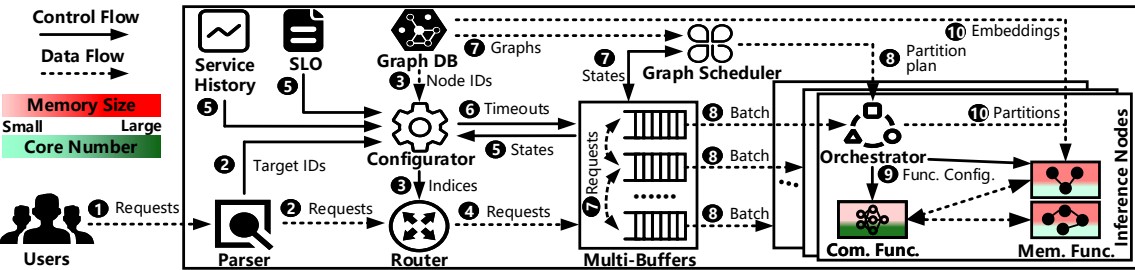

**Figure 7: A system overview of $\lambda$Grapher.**

and memory redundancy, and (2) resource-centric function management which involves fine-grained resource allocation for functions in the form of compute function groups and memory function groups, catering to the compute-sensitive and memory-sensitive operations, thereby maximizing resource efficiency. $\lambda$Grapher aims to optimize the resource efficiency during GNN serving while ensuring the latency SLOs of GNN requests.

Figure 7 illustrates the system overview of $\lambda$Grapher. At the beginning of the GNN serving, ❶ a continuous stream of user requests arrives at the serverless platform. Then, ❷ the *Parser* analyzes the content of the user requests, i.e., the IDs of the target vertices that require processing on the graph, and the target IDs are dispatched to the *Configurator*, while the user requests enter the *Router*. Next, ❸ the *Configurator*, based on the received target IDs, queries the vertex IDs within an $n$-hop computation graph around the target IDs from the graph database (excluding the graph object and feature vectors) and generates the corresponding data indices to send to the *Router*. According to the data indices, ❹ the *Router* routes incoming user requests to the buffer with the highest degree of graph sharing among the *Multi-Buffers*. While the requests are waiting in the *Multi-Buffers*, ❺ the *Configurator* collects the states of the buffers and queries the built-in latency SLO as well as the historical service logs to ❻ periodically set and adjust the timeouts for each buffer in the *Multi-Buffers*. As the requests continue to be added to the *Multi-Buffers*, ❼ the *Graph Scheduler*, with a global perspective, schedules the requests within the *Multi-Buffers* based on the state of each buffer (i.e., scheduling requests to move in or out of buffers) to enhance the benefits of graph sharing, and extracts $n$-hop computation graphs corresponding to the requests from the graph database, performing dynamic graph partitioning on each buffer. When a buffer times out, ❽ the batched requests and results are sent to the newly created *Orchestrator*. The *Orchestrator*, based on the graph partitions, ❾ scales the compute resource function groups and memory resource function groups, mapping the workloads to specific functions. During the runtime loading process, ❿ the compute functions load only the neural network structure, while the memory functions load the graph structure of partitions and its corresponding feature vectors. Finally, the functions perform collaborative inference as per the orchestrated process. Next, we present the details of each module.

## 3.2 Parser and Router

When user requests continuously arrive at the serverless platform, it is necessary to analyze relevant information from the requests for subsequent inference and route the requests to designated buffers for graph sharing with other requests.

**Target IDs.** The *Parser* is responsible for analyzing the content of user requests, which are the IDs of the target graph structures that need to be inferred, referred to as the target IDs. Graph analysis tasks can primarily be categorized into three types: vertex-level prediction, edge-level prediction, and graph-level prediction. Taking social network analysis as an example, vertex-level prediction involves determining user interests, with the target IDs for requests being the vertex IDs. Edge-level prediction involves analyzing relationships between users, and the target IDs for requests are the edge IDs. Graph-level prediction pertains to the overall properties of specific information within the entire social network, with the target IDs for requests being the computation graph IDs. Therefore, the *Parser* needs to analyze essential information from requests based on the type of analysis task to support the subsequent extraction of the $n$-hop computation graphs.

**Routing Strategy.** The *Router* is responsible for routing each request to the buffer that has the highest graph-sharing degree for that request to enhance data locality. Each request corresponds to an $n$-hop computation graph $G(V, E)$ based on its task ID. The configurator, upon receiving the task ID, extracts the $n$-hop computation graph from the graph database that stores the whole graph for the service and generates the data index for that request $r_i$ by using the vertex set $V$ of this computation graph as $U_{r_i} = V_{r_i}$, where $U_{r_i}$ is the data index of the request $r_i$. The data index for a buffer $b_i$ is the union of data indices for all requests it contains:

$$U_{b_i} = U_{r_0} \cup U_{r_1} \cup \ldots \cup U_{r_j}, r_j \in b_i. \tag{3}$$

The routing strategy involves directing requests to the buffer with the highest graph-sharing degree, which is determined by finding the intersection between the data index for each buffer and the request's data index, with the largest intersection indicating the highest graph-sharing degree buffer:

$$S_{b_i}^{r_i} = \left| U_{b_i} \cap U_{r_i} \right| / \left| U_{r_i} \right|, b_j = \arg\max_B S_B^{r_i}, \tag{4}$$

where $S_{b_i}^{r_i}$ is the graph sharing degree of request $r_i$ with respect to buffer $b_i$ in Multi-Buffers $B$, and $b_j$ is the buffer with the highest graph sharing degree for the request $r_i$.

## 3.3 Multi-Buffers and Configurator

**Multi-Buffers.** We observe a significant overlap among the computation graphs corresponding to user requests arriving continuously over a period as discussed in Section 2.2. Therefore, we design the *Multi-Buffers* which provides requests with an opportunity for graph sharing with other requests with the same part of the computation graph, by allowing requests to wait in the buffer for a certain period. The *Multi-Buffers*, denoted as $B$, consists of multiple

individual buffers. The requests that can engage in graph sharing are placed in the same buffer and continue to wait for other requests eligible for graph sharing. The requests are processed and sent to subsequent inference nodes for inference only when the current buffer times out. Each buffer possesses a 5-tuple $(R, S, N, Q, K)$ to characterize the state of the buffer at the current moment, where $R$ denotes the requests per second for the buffer, $S \in [0, 1]$ represents the average graph sharing degree of all requests in the buffer, $N$ represents the number of the requests in the buffer, $Q \in [0, 1]$ indicates the ratio between the remaining time and the configured timeout of the buffer, and $K \in [0, 1]$ represents the ratio between the buffer's configured timeout and the maximum allowable timeout setting. The timeout of the buffer is a crucial determinant of system performance. Configuring the appropriate timeout enables efficient resource conservation through graph sharing while simultaneously ensuring the timely fulfillment of SLO requirements. However, setting the timeout too high or too low can result in request violations or diminished graph-sharing benefits, reducing system resource efficiency. In the evolving inference service environment, we need to configure the buffer's timeout reasonably.

**Adaptive Timeout Configuration.** The *Configurator* adaptively configures the buffer's timeout based on the buffer's state to balance the benefits of graph sharing and the timeliness of inference. We utilize the decision tree regression algorithm [43] to capture the buffer state and make rapid and effective timeout adjustment decisions, ultimately achieving a balanced benefit. We employ real-world traces from Twitter [2, 7] and utilize the built-in SLO to conduct authentic service runs, thereby gaining service history:

**Step 1:** we determine the initial timeout $T_0$ and the maximum timeout $T_{max}$ for the buffer Based on the SLO:

$$T_0 = \gamma \times SLO, \ T_{max} = \delta \times SLO, \ 0 < \gamma < \delta < 1. \quad (5)$$

**Step 2:** Whenever the buffer accumulates a certain number of new requests or after a certain period has passed, the buffer's state changes and a decision needs to be made from the decision set $X = \{-1, 0\} \cup \{1 \times \tau, \dots, i \times \tau\}$. There are three types of decisions, including $x_i = -1$, indicating an immediate sending of the buffer to the inference node for the execution of the inference phase, $x_i = 0$, indicating the preservation of the existing timeout, and the extension of the timeout by $x_i = i \times \tau$, where $\tau$ is the unit time interval for extending the timeout. To assess the magnitude of each decision's benefits, we propose a metric that measures the trade-off between graph-sharing benefits and inference timeliness:

$$\mu_{b_i} = \alpha \times S_{b_i} - \beta \times D_{b_i}, \alpha \in [0, 1], \beta \in [0, 1], \quad (6)$$

where $\mu_{b_i}$ represents the total performance gain of the buffer $b_i$, $S_{b_i}$ signifies the benefit of graph sharing obtained in the buffer $b_i$, which is the average graph sharing degree, $D_{b_i}$ denotes the average time ratio delayed in the buffer $b_i$ due to waiting, i.e., the average of the time each request is delayed in the buffer relative to the Timeout, and $\alpha$ and $\beta$ are fixed coefficients set by the developers. We record the buffer's state and the decision with the maximum performance gain when a decision is required, using this as historical experience. **Step 3:** Using the 5-tuple state of the buffer $(R, S, N, Q, K)$ as independent variables and the corresponding decisions as the dependent variable, we fit a decision tree regression model:

$$Model = DecisionTreeRegressor.fit((R, S, N, Q, K), X). \quad (7)$$

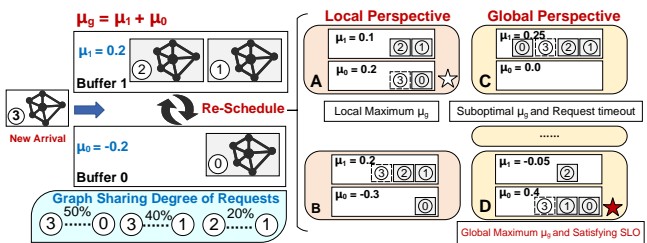

**Figure 8: The global perspective optimization process.**

The generated *Configurator* based on the decision tree regression model brings us some decision-making heuristics: 1) When the buffer has been waiting for a long time without the prospect of achieving graph sharing or when the graph sharing degree is sufficiently high, the *Configurator* tends to decide $x_i = -1$, opting not to continue waiting and instead sending the buffer to subsequent inference node; 2) When the benefits remain stable, the *Configurator* tends to prefer the choice of $x_i = 0$, conservatively maintaining the current timeout; 3) When the buffer consistently receives requests that can significantly enhance the average graph sharing degree, the *Configurator* tends to decide of $x_i = i \times \tau$, which involves greedily extending the timeout, with the degree of greediness depending on the extent of benefit increase.

### 3.4 Graph Scheduler

The *Graph Scheduler* is responsible for overall scheduling of the computation graphs corresponding to the requests in the *Multi-Buffers*, which involves three specific parts: 1) Globally adjust requests between the buffers to achieve the optimal scenario for graph sharing; 2) Conduct graph sharing by merging common vertices in the computation graphs to reducing computation and memory redundancy; 3) Dynamically partition the request graphs to enhance resource efficiency and provide scalability for inference.

**Global Perspective Optimization.** When a new request arrives, it is always routed to the buffer with the highest graph sharing degree for itself according to the routing strategy. However, this can lead to the convergence of graph sharing results towards local optima rather than global optima, as shown in Figure 8. Therefore, we introduce the global perspective optimization algorithm to dynamically and adaptively schedule the remaining requests with graph sharing from a global perspective to achieve the global optimum of graph sharing, as demonstrated in Algorithm 1 in Appendix B. Whenever a new request enters the *Multi-Buffers*, the *Graph Scheduler* places this request in the appropriate buffer based on the routing strategy and calculates the current buffer's performance gain (Line 1-Line 3). The *Graph Scheduler* detects other requests in other buffers that can participate in graph sharing with the newly arrived request and calculates their respective graph sharing degree (Line 4-Line 7). Next, the *Graph Scheduler* calculates the average graph sharing degree and the average time ratio delayed if requests are moved in or out of the buffer, thus computing the performance gain after scheduling (Line 8-Line 12). Finally, the *Graph Scheduler* compares the performance gains before and after the above scheduling. If the performance gain is greater after the scheduling, the *Graph Scheduler* adopts this decision, transferring the corresponding requests into the buffer where the new request is located and removing

Figure 9: Demonstration of the dynamic graph scheduling.

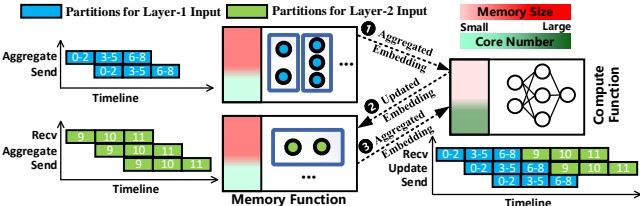

Figure 10: Demonstration of the collaborative inference.

them from their respective buffer (Line 13-Line 16). After scheduling from a global perspective, the requests in the buffer are the global optimal solution for graph sharing. After scheduling from a global perspective, requests in the buffer can make full use of the benefits brought by subsequent graph sharing.

**Graph Sharing.** The *Graph Scheduler* combines the graphs corresponding to all requests in the buffer to reuse the Intermediate results and reduce computational and memory redundancy in subsequent batch processing. Specifically, the *Graph Scheduler* adopts a hierarchically aggregated computation graph (HAG), based on the ideas from previous work [14], to merge redundant vertices and facilitate result sharing on the graph. The process of graph sharing primarily involves three steps: 1) Expand the computation graph of the target vertex into a computation tree; 2) Traversal the computation tree to merge vertices at the same depth between different computation trees; 3) Conduct the aggregation operation on the merged vertices only once, and the intermediate results from the aggregation operation can be reused in subsequent steps. Taking Figure 5a as an example, two requests have computation trees that simultaneously share 5 overlap vertices. These overlapping vertices are first merged and aggregated to produce intermediate results, which are then further aggregated with other context vertices to complete the inference of the target vertex. Compared to executing the two computation trees separately, graph sharing significantly reduces the number of aggregation operations and data transfers.

**Dynamic Graph Scheduling.** In the *Graph Scheduler*, each buffer corresponds to a large graph composed of requests. The *Graph Scheduler* performs graph scheduling on a per-buffer basis. Whenever a new request is added to a buffer, the *Graph Scheduler* extracts the computation graph structure of that request from the graph database. Similarly, when requests are transferred between buffers, the corresponding graph structures are also transferred. This dynamic incremental graph partitioning is carried out to optimize inference for subsequent tasks. The dynamic graph scheduling serves three main objectives, as shown in Figure 9: 1) Not all graph vertices participate in the computation at every layer, as illustrated in Figure 2c. Dynamic graph scheduling involves partitioning the graph for each GNN layer, leveraging the resource-efficient nature of serverless functions that are created and destroyed as needed; 2) Serverless function instances have resource limitations and cannot accommodate the entirety of the graphs stored in buffers. Dynamic graph scheduling provides scalability for inference, addressing this constraint. Algorithm 2 in Appendix B describes the specific dynamic graph partitioning process. First, combine the graphs in the buffer with the arrived request graph to generate the HAG, which is the data structure resulting from shared graph scheduling (Line 1). Next, begin traversing from the task vertex to its predecessor vertices (note that even in the case of an undirected graph, it is

represented as a directed graph), i.e., the vertices required for its aggregation, which are formed as a partition (Line 2-Line 9). The predecessor vertices visited in the previous iteration are treated as new task vertices for the subsequent traversal, and this process continues until the set of task vertices becomes empty, at which point the algorithm concludes (Line 10-Line 12). Finally, we obtain a two-dimensional list of graph partitions, where each row represents the input for each GNN layer, and the granularity of these graph partitions is fine, providing scalability for subsequent inference.

## 3.5 Orchestrator

The *Orchestrator* coordinates a set of serverless functions to perform GNN inference on batched requests, as shown in Figure 10, following resource-centric management that maximizes resource efficiency without violating SLO, which comprises three stages: 1) The *Orchestrator* maps memory-sensitive graph workloads and compute-sensitive tensor workloads to memory functions and compute functions, respectively; 2) The *Orchestrator* employs a pipeline collaborative inference mechanism to distribute communication overhead among functions; 3) Based on the workloads and the remaining time, the *Orchestrator* scales memory functions and compute functions, customizing their resource allocation.

**Workload Mapping.** The *Orchestrator* divides the GNN workload into graph workloads and tensor workloads and manages serverless functions with resource groups, categorized into memory function groups with abundant memory resources and compute function groups with ample computing resources. The memory function group exclusively handles graph workloads, i.e., memory-sensitive Aggregate operations, while the compute function group exclusively loads tensor workloads and handles computation-sensitive Update operations. For the mapping of graph workloads, to leverage the resource-efficient nature of serverless functions and save resources, the *Orchestrator* maps different layers of GNN's input graph partitions to different memory functions, and maps partitions from the same GNN layer to the same memory function whenever possible. If the memory size of partitions from the same GNN layer exceeds the instance memory limit, the excess partitions will be mapped to another new memory function instance. Regarding the mapping of tensor workloads, as tensor workloads require less memory, the neural networks for each GNN layer are loaded into a single compute function instance. The target vertices that complete their tasks early can exit the batch processing and return the results.

**Collaboration between Functions.** The *Orchestrator* organizes collaborative inference between functions in a pipeline fashion, allowing the communication overhead between functions to be distributed within their respective computations, as illustrated in Figure 10. The entire pipeline process begins with the memory function inferring the first layer of GNN, and thus, the granularity

of concurrent tasks in the pipeline is determined by the number of graph partitions and vertices processed in parallel at each step by the first layer memory function. The concurrent granularity needs to be considered when allocating resources for functions.

**Function Scaling.** The *Orchestrator* customizes resources for memory functions and compute functions based on workload size and concurrent granularity, saving resources while ensuring SLO compliance. Specifically, the allocated memory resource amount for memory function $F_i^m$ and compute function $F_i^c$ are $M_i^m$ and $M_i^c$:

$$M_i^m = M_{runtime} + M_{V_i} + M_{E_i} + M_{h_i} \tag{8}$$

$$M_i^c = M_{runtime} + M_{nn} \tag{9}$$

where $M_{runtime}$ represents the runtime memory size, $M_{V_i}$ represents the memory size of loaded vertices, $M_{E_i}$ represents the memory size of loaded edges, $M_{h_i}$ represents the memory size of loaded embeddings, and $M_{nn}$ represents the memory size of the neural network. the *Orchestrator* allocates the CPU cores to functions based the bayesian optimization [32]:

$$BayesianOptimization(\vec{F}, \vec{X}, \vec{\Gamma}, \vec{T_l}) \rightarrow \vec{C} \tag{10}$$

$$Minimize: Cost = \sum F^m \times T_l^m + \sum F^c \times T_l^c \tag{11}$$

$$Constrains: \sum T_l^m + \sum T_l^c \leq T_{SLO} - T_{timeout} \tag{12}$$

where $\vec{F}$ represents the function vector, $\vec{X}$ represents the function workload size vector, $\vec{\Gamma}$ indecates the concurrent granularity vector, $\vec{T_l}$ represents the inference time vector under different cores and task size, $\vec{C}$ represents the core number vector, and $T_l^m$ and $T_l^c$ indicate the inference time of memory and compute functions.

## 4 EVALUATION

In this section, we prototype λGrapher and evaluate it with real-world traces from various web applications.

### 4.1 Experimental Setup

**λGrapher Prototype.** We prototype λGrapher based on the open-source serverless platform Knative [19] with 3.5k LOC in Python and Go. Specifically, we implement the *Parser, Configurator, Router, Multi-Buffers, Graph Scheduler*, and *Orchestrator* in a VM instance as intermediaries between the request source and the Knative platform and we deploy function instances through Knative Serving Service and Knative Serving Ingress. The graph query service is implemented using the high-performance Neo4j [39] graph database.

**Baselines.** We compare λGrapher with two state-of-the-art GNN serving systems, including GraphLearn [1], representing the traditional cloud service architecture, and a financial fraud detection system based on AWS Lambda [6], donated as AWSGNN, representing the request-centric serverless architecture. GraphLearn relies on monitoring memory occupancy threshold (80%) metrics to scale instances (1GB, 8vCPU) up or down, as most traditional elastic cloud services do [4]. AWSGNN dynamically allocates functions for each request based on its requirements, with a fixed ratio of memory to computational resources (128MB to 1vCPU) [8].

**Web Application Traces.** We utilize real-world traces from Twitter [2] to generate the inter-arrival time of user requests, which is widely used for evaluating inference systems. We use three graph datasets from real-world applications to generate request

**Table 1: Graph Datasets from Real-World Applications**

| Graph Datasets | Graph Type | \|V\| | \|E\| | Dim. | SLO (s) |
|---|---|---|---|---|---|
| Bitcoin OTC [18] | Unipartite, Directed | 5,881 | 35,592 | 128 | 0.3 |
| KuaiRec [10] | Bipartite, Undirected | 17,904 | 192,729 | 58 | 0.4 |
| Higgs Twitter [11] | Unipartite, Directed | 456,626 | 14,855,842 | 128 | 0.6 |

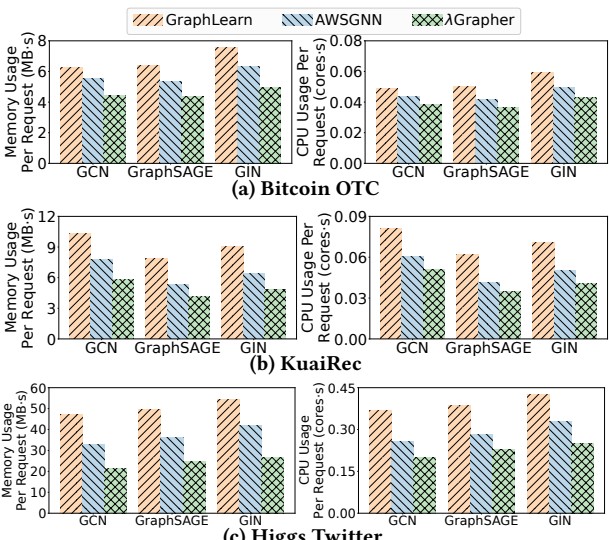

**Figure 11: Resource efficiency between λGrapher and two state-of-the-art under different traces and GNN workloads.**

contents, including KuaiRec [10] from the video-sharing mobile app Kuaishou [20], Bitcoin OTC [22] form Bitcoin transaction network, and Higgs Twitter [7] from Twitter network, which are widely applied in evaluating the GNN model designed for short-video recommendation [27], financial fraud detection [21] and social network analysis [31], respectively. The SLOs are set based on the requirements of the application scenario, as described in the previous work [48]. The details of graph datasets are shown in Table 1.

**GNN Workloads.** We select three common GNN models with three layers using the Deep Graph Library (DGL) [37], including GCN [18], GraphSAGE [11], and GIN [42]. The structures of GNN layers are shown in Figure 13 in Appendix A.

**Testbed.** We implement λGrapher on a local cluster with 10 physical machines, each of which includes 104 Intel Xeon 8269CY cores at 2.5GHz and 192 GB RAM (Ubuntu 18.04). We collect real service data on physical machines, such as inference latency under various configurations. To expedite the experimental process, we transform the prototype implementation into a simulation mode as in [24].

### 4.2 Performance Comparison

We compare λGrapher with the two state-of-the-art GNN serving systems, GraphLearn and AWSGNN, in terms of memory and compute resource efficiency, specifically comparing the average memory and compute resource usage per request. The results in Figure 11 indicate that, compared to the state-of-the-art, λGrapher can achieve up to 54.2% memory and 45.3% in computing resource savings. Across three web application traces, including Bitcoin OTC, KuaiRec, and Higgs Twitter, the average graph sharing degrees of each buffer are 44.8%, 50.6%, and 61.8% respectively.

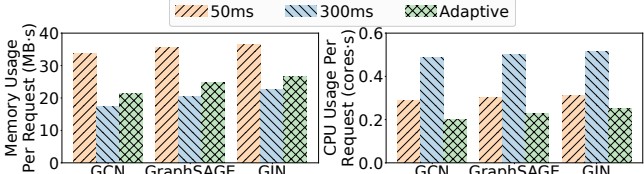

**Figure 12: Results of the adaptive timeout configuration.**

**Memory Resource Efficiency.** In different GNN workloads and across various traces, $\lambda$Grapher reduces memory resource usage by 28.9% to 54.2% compared to GraphLearn and 18.7% to 36.7% compared to AWSGNN (on average 34.6%). As a representative of traditional cloud-based systems, GraphLearn still employs an approach of over-allocating resources prior to service initiation and continuously monitoring to address request fluctuations and system availability. However, its instance resource scaling granularity is relatively coarse, resulting in significant memory resource wastage. As a representative of request-centric serverless systems, AWSGNN effectively manages request intensity fluctuations. Nonetheless, due to significant overlaps between requests arriving in close proximity during the same time frame, the approach of serving individual requests with individual functions lacks the utilization of spatial data locality, leading to memory redundancy during GNN inference. $\lambda$Grapher adopts a graph-centric task scheduling approach, which efficiently saves memory resources by scheduling requests that can perform graph sharing together through adaptive buffer timeout configuration and global perspective request scheduling.

**Computing Resource Efficiency.** Under various GNN workloads and across different traces, $\lambda$Grapher demonstrates a reduction in computing resource usage, achieving savings ranging from 21.5% to 45.3% compared to GraphLearn, and 12.1% to 23.5% compared to AWSGNN (on average 26.7%). As illustrated in Figure 3, GNN's fine-grained operations exhibit significantly different resource sensitivities. However, in both GraphLearn and AWSGNN, resource allocation is coarse-grained for the entire GNN, resulting in suboptimal utilization of computing resources. $\lambda$Grapher not only reduces unnecessary computational redundancy through graph sharing but also offers a resource-centric function management mechanism. By decoupling Aggregate and Update operations with varying resource sensitivities, $\lambda$Grapher enables fine-grained resource allocation. Additionally, $\lambda$Grapher orchestrates a refined pipeline for customized functions to ensure load balancing. These improvements lead to a substantial enhancement in computing resource efficiency.

### 4.3 Sensitivity Analysis of Buffer Timeout

To validate the performance of the adaptive timeout configuration module, we select fixed upper and lower bounds for buffer timeout, set at 50ms and 300ms, respectively. $\lambda$Grapher dynamically adjusts within this range. We conduct tests on the largest-scale graph datasets Higgs Twitter, under various GNN workloads, as shown in Figure 12. The results demonstrate that the adaptive timeout configuration scheme can save 31.2% of memory resources and 24.9% of computational resources on average compared to a fixed configuration with a lower limit of 50ms. Compared to the upper limit configuration 300ms, it achieves an average 54.8% reduction in computing resource usage. The underlying reason for this is

that, with a 50ms timeout, it cannot leverage data locality between requests to optimize resource efficiency. Conversely, with a 300ms timeout, although it can fully harness graph sharing to optimize memory resource efficiency, the extended request waiting times require a significant amount of computational resources to ensure compliance with SLO goals. The adaptive timeout configuration dynamically adjusts the timeout size based on the buffer status, providing a balance between the benefits of graph sharing and the risk of violating SLO.

## 5 RELATED WORK

**GNN Inference.** In the traditional distributed environment, the focus of work in recent years is how to divide the graph reasonably and map fine-grained operations to computing resources of appropriate size to achieve acceleration [3, 17, 38, 49]. Wang et al. [38] propose an adaptive and efficient system for GNN acceleration on GPUs, which preprocesses the model and input graph to achieve reasonable graph partitioning and resource mapping, and finally achieves accelerated inference. In the cloud environment, in order to solve the problem of graph data distributed in different geographies, Zeng et al. [45] propose to conduct the GNN real-time inference by adopting the fog computing paradigm to reduce the communication overhead of the data collection before inference. The above works focus on inference of static GNN models, which pre-allocate computing node resources and provide services by continuous monitoring. This scheme is difficult to dynamically and adaptively allocate resources according to the fluctuation of user requests, resulting in waste of resources.

**Serverless Graph System.** Due to the elastic scalability and flexibility of serverless computing, some scholars propose to migrate the graph processing system to the FaaS platform in recent years [12, 33, 34]. Toader et al. [34] implement the classic large-graph processing model Pregel [28] on the FaaS platform in a simple engineering manner, and introduce a remote storage mechanism to meet the stateless challenge. However, due to frequent data communication, the system performs poorly in performing large-scale graph algorithms. Thorpe et al. [33] make the GNN training process semi-serverless, introducing serverless threads to handle computation-sensitive tensor operations, while graph operations that are sensitive to memory resources are still executed on the CPU server. At present, there is a gap in the work of serverless-based GNN serving.

## 6 CONCLUSION

In this paper, we identify the resource inefficiency problem in current GNN serving systems. Through studying the web application traces, we observe the spatial data locality in computation graphs of requests. We propose a scalable, resource-efficient serverless system named $\lambda$Grapher for GNN serving. $\lambda$Grapher supports a graph-centric task scheduling strategy to reduce the computation and memory redundancy and facilitates a resource-centric function management mechanism which allocates resources to functions catering to the resource sensitivities of GNN fine-grained operations. Compared to the state of the arts, our $\lambda$Grapher prototype can save up to 54.2% in memory resource usage and 45.3% in computing resource usage with real-world traces while meeting the SLOs.

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

## A GNN LAYER STRUCTURES

Figure 13 shows the the structure of three classic GNN layers, namely GCN [18], GraphSAGE [11], and GIN [42]. Each GNN layer is composed of two main operations alternatively executed: Aggregate and Update.

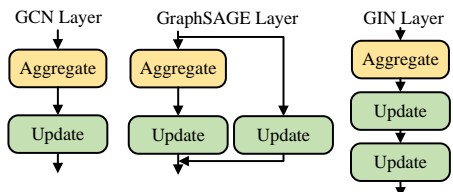

**Figure 13: Classic GNN Layers.**

## B ALGORITHM DETAILS

---

**Algorithm 1:** Global Perspective Optimization Algorithm

| | |
|---|---|
| **Input** | : Multi-Buffers $B = \{b_0, b_1, \ldots, b_i\}$; |
| | Requests in the buffer $b_i = \{r_0^{b_i}, r_1^{b_i}, \ldots, r_i^{b_i}\}$ ; |
| | Arrived requests $A = \{r_0, r_1, \ldots, r_k\}$; |
| | $A$'s routing IDs $J = \{j_{r_0}, j_{r_1}, \ldots, j_{r_i}\}$; |
| **Output** | : Modified Multi-Buffers $B'$; |
| **Parameters** | : Graph sharing degree of request with buffer $S_b^r$; Buffer to which the request is routed $b_j$; Data index of request $U_r$; Average sharing degree of buffer $S_b, S_b'$; Average time ratio delayed in the buffer $D_b, D_b'$; Performance gain of buffer $\mu_b, \mu_b'$; Fixed coefficients $\alpha, \beta$; |

1 **foreach** $r_k$ in $A$ **do**
2    routeRequestToBuf($r_k, j_{r_k}$) $\rightarrow b_j$;
3    $\alpha \times S_{b_j} - \beta \times D_{b_j} \rightarrow \mu_{b_j}$;
4    **foreach** $b_i$ **do**
5      **if** $b_i \neq b_j$ and $S_{b_i}^{r_k} > 0$ **then**
6        **foreach** $r_i$ in $b_i$ **do**
7          $S_{r_i}^{r_k} = \frac{|U_{r_k} \cap U_{r_i}|}{|U_{r_k}|}$;
8          **if** $S_{r_i}^{r_k} > 0$ **then**
9            modBufByDelRequest($b_i, r_i$) $\rightarrow S_{b_i}', D_{b_i}'$;
10            $\alpha \times S_{b_i}' - \beta \times D_{b_i}' \rightarrow \mu_{b_i}'$;
11            modBufByAddRequest($b_j, r_i$) $\rightarrow S_{b_j}', D_{b_j}'$;
12            $\alpha \times S_{b_j}' - \beta \times D_{b_j}' \rightarrow \mu_{b_j}'$;
13            **if** $\mu_{b_i}' + \mu_{b_j}' > \mu_{b_i} + \mu_{b_j}$ **then**
14              transferRequestToBuf($r_i$) $\rightarrow b_j$;
15              delRequestFromBuf($r_i, b_i$);
16 **return** $B'$;

---

**Algorithm 2:** Dynamic Graph Scheduling Algorithm

| | |
|---|---|
| **Input** | : Graph of the buffer $G_b(V_b, E_b)$; |
| | Graph of the arrived request $G_r(V_r, E_r)$; |
| | Target vertices IDs $W = [w_0, \ldots, w_i]$ ; |
| **Output** | : Graph Partitions for each GNN layer $P = [[p_{00}, \ldots, p_{0i}], \ldots, [p_{n0}, \ldots, p_{ni}]]$ |
| **Parameters** | : HAG $H(V_h, E_h)$; Traverse depth $n$; |

1 GenerateHAG($G_b(V_b, E_b), G_r(V_r, E_r)$) $\rightarrow H(V_h, E_h)$;
2 $[] \rightarrow P$ and $0 \rightarrow n$;
3 **while** $W \neq \varnothing$ **do**
4    $p_n = []$;
5    **foreach** $w_i$ in $W$ **do**
6      traversePredecessors($w_i, H(V_h, E_h)$) $\rightarrow p_{ni}$;
7      append($p_{ni}$) $\rightarrow p_n$;
8    append($p_n$) $\rightarrow p$;
9    $p_{n0} \cup p_{n1} \cup \ldots \cup p_{ni} \rightarrow W$;
10    $n + 1 \rightarrow n$;
11 **return** $P$;

---

