# OpenReview forum: "𝜆Grapher: A Resource-Efficient Serverless System for GNN Serving through Graph Sharing"
_ACM.org/TheWebConf/2024/Conference — TheWebConf24 Oral_

### Official Review · Reviewer_CeKB · 2023-11-07

**Novelty:** 6
**Technical Quality:** 5

**Review:**

# Quality

+ The paper seems to be of high quality, grounded in extensive empirical analysis.
+ The authors' approach to addressing GNN serving challenges is systematic and thorough, with well-defined problem statements and solutions.

# Clarity

+ The paper provides a clear problem statement and the proposed solution, making it accessible to readers.
+ The paper promises easy-to-follow sections that should logically lay out the problem, proposed solution, evaluation, related work, and conclusion.

# Originality

+ The idea of leveraging data locality in serverless computing for GNN serving is original and demonstrates a novel approach to addressing resource inefficiency.
+ The combination of adaptive timeout, graph-centric scheduling, and resource-centric function management contributes to the field.

# Significance

+ Scalability in GNN serving is a critical issue, and the paper's contributions could significantly impact web services relying on GNNs.
+The proposed system potentially offers substantial resource savings, which is highly significant for operational costs and efficiency.

# Pros
+ The paper's core idea of leveraging data locality and graph-centric scheduling for resource efficiency is intuitive and insightful.
+ The challenges of GNN serving are well-articulated, making the motivation behind the method clear and compelling.
+ Addressing scalability reflects an understanding of practical deployment concerns, and the achieved resource savings indicate a significant step forward.
+ The structure of the paper suggests a coherent presentation, which is likely to facilitate comprehension among readers.

# Cons
- The paper does not discuss potential barriers to adoption, such as the need for companies to adjust their existing serverless architectures.
- It may require renovation when the data locality is less useful as data are evolving.
- Some minor typos and grammar errors.
- Limited discussion with existing serverless Graph Systems in Section 5, Related Work.

**Questions:**

- Though the evaluation results can show efficiency improvement, but I'm wondering whether it will influence the accuracy of GNN-based computing. If not, how can you show that?
- If the data distribution shifts and the current data locality changes, how can your approach catch such an evolution?

**Reviewer Confidence:**

3: The reviewer is confident but not certain that the evaluation is correct

**Scope:**

4: The work is relevant to the Web and to the track, and is of broad interest to the community

---

### Official Review · Reviewer_q5j1 · 2023-11-17

**Novelty:** 4
**Technical Quality:** 5

**Review:**

The paper proposes Grapher, a serverless system for GNN serving that achieves resource efficiency through graph sharing and fine-grained resource allocation, which features adaptive timeout for request buffering, graph-centric scheduling, and resource-centric function management with fine-grained resource allocation catered to the resource sensitivities of GNN operations and function orchestration optimized to hide communication latency.

Pros:
1.	Grapher achieves resource efficiency through graph sharing and fine-grained resource allocation, leading to significant savings in memory and computing resources compared to state-of-the-art systems.
2.	The system features an adaptive timeout mechanism for request buffering, which can enhance the overall performance and responsiveness of the system.
3.	Grapher utilizes a graph-centric request scheduler, which can optimize the allocation of resources based on the characteristics of the graph, potentially improving the overall efficiency of GNN serving.
4.	The authors present an empirical analysis of GNN workload variations, data locality, and resource sensitivities of GNN operations, providing a solid foundation for the proposed method.

Cons:
1.	Implementing and managing Grapher seems to introduce complexity in terms of configuration, maintenance, and potential troubleshooting, the author should provide its practicality.
2.	The system's resource-centric function management with fine-grained resource allocation may introduce challenges related to managing and optimizing resource allocation for different GNN operations, potentially requiring careful tuning and monitoring
3.	The adaptive timeout configuration, while beneficial for balancing graph sharing benefits and timeliness of inference, may introduce complexities in determining the appropriate timeout settings, potentially impacting system performance if not configured optimally

**Questions:**

please answer the questions above.

**Reviewer Confidence:**

2: The reviewer is willing to defend the evaluation, but it is likely that the reviewer did not understand parts of the paper

**Scope:**

3: The work is somewhat relevant to the Web and to the track, and is of narrow interest to a sub-community

---

### Official Review · Reviewer_sjeq · 2023-11-21

**Novelty:** 5
**Technical Quality:** 4

**Review:**

The paper aims to improve the resource efficiency of the GNN Serving platform. The authors observe the significant data locality in computation graphs of requests and propose an efficient method to improve resource efficiency from a graph-level perspective. The key designs of the paper consist of (1) graph-centric scheduling through graph sharing; and (2) resource-centric function management with fine-grained resource allocation catered to the resource sensitivities of GNN operations. The methods have a notable improvement compared to traditional request-level methods. The paper implements a prototype on the open-source serverless platform Knative and evaluates it with real-world traces from various web applications. The results prove the memory resource efficiency and computing Resource Efficiency of the proposed methods.

Strengths:
	This paper studies an important problem to improve the resource efficiency of GNN serverless platform.
	The proposed method exploits the operator sharing among different requests to minimize computation and memory redundancy.
	The authors decouple GNN operations for fine-grained resource allocation.

Weaknesses:

	The overhead of exploiting the sharing graph is unclear.
	Lack of ablation experiments.
	The impact of the proposed methods on the computation time of requests remains unclear.
	The motivated example for data locality and graph sharing is confusing.
	The graph sharing (line 601-607) lacks finer design details.

Detailed comments:

1、Graph sharing is the key technology of this paper. However, the experimental part did not see the overhead evaluation of graph sharing. It should be pointed out that when graph sharing features between requests are sparse, graph sharing seems to become a performance bottleneck.

2、The paper should increase the evaluation of each key technology component.

3、The impact of the proposed methods on the computation time of requests remains unclear. The computation time of functions is a critical metric in the real-world serverless computing platform.

4、The concepts of request-level, graph-level, and layer-level, especially Fig. 2(b), are confusing for me. Additionally, the relationship between the observation “data locality” and the core contribution of this paper, graph-sharing, is somewhat obscure and difficult to understand.

5、The steps of prior work HAG are already clear, which is presented in line 601-607. However, the process of exploiting the sharing graph between different requests has not yet been presented in detail, e.g. the design of the number of vertices and requests.

**Questions:**

1、What is the relationship between the proposed graph sharing and autoscaling?

2、Provide a more detailed description of graph sharing and analyze its overhead.

3、Analyze the potential additional computation time overhead that the proposed method may incur.

4、Clarify how the observation of "data locality" serves as motivation for the graph-sharing method.

**Reviewer Confidence:**

3: The reviewer is confident but not certain that the evaluation is correct

**Scope:**

2: The connection to the Web is incidental, e.g., use of Web data or API

---

### Official Review · Reviewer_t83Z · 2023-11-22

**Novelty:** 4
**Technical Quality:** 5

**Review:**

Summary:

This paper proposes a new scheduler for GNN inference requests.  Specifically, the paper proposes to batch incoming requests and schedule batches of requests using a serverless computing framework.  The goal here is to reduce the memory and compute footprint of the workload by loading one subset of the graph into memory and using it to serve an entire batch of requests.  This avoids redundantly loading the same parts of the graph into memory to serve requests that access similar parts of the graph.  Some computation over these shared vertices and edges can also be reused between similar requests.  The problem, then, is to both figure out how to maximize these savings while also maintaining acceptable performance.  Answering this question involves deciding how to sort requests into batches that access similar parts of the graph, how long to hold each batch before serving the requests in the batch, and how to serve a batch of similar requests so as to minimize compute and memory utilization.

The paper provides solutions to these questions and then evaluates the performance of the solution against two other popular GNN frameworks in a real system serving requests from 3 real-world graph inference workloads.

Pros

The writing was clear and descriptive

The problem considered is a relevant, interesting, hard problem.  The proposed solution clearly decomposes the problem into several interesting subproblems and proposes a reasonable solution for each subproblem

The proposed solution is evaluated using a real-world implementation on reasonable workloads and compares favorably to the state-of-the-art

Cons

Seemingly due to space constraints the depth of the evaluation is somewhat shallow.

It is somewhat hard to tell which features in the proposed solution are the most effective and important for reducing resource utilization

A more in-depth evaluation of request latencies and general system overheads would also be helpful

Overall:

In general, this is a very nice paper and I recommend accepting it.  My feedback is mostly geared towards improvements for the final version of the paper.  The system description section is quite long and the evaluation section is quite brief and does not provide detailed breakdowns of system performance and overheads.  Additional results here would be very interesting to see.  I would recommend moving some of the lower level details of the implementation to an appendix or describing them a higher level to make space for a more detailed evaluation.

**Questions:**

I do have some questions about the prior work.  It seems that there is some work both on caching for serverless computing and caching for graph-based computation.  There are algorithms/policies for caching in both of these cases.  Does this kind of work apply here, and if not why does your cache partitioning strategy work better than the state-of-the art?

How do the SLO violation rates compare between the proposed solution and the existing solutions?  Can you show a graph describing the tradeoff between delay and resource usage?

What are the various overheads in the system and how do they compare to existing solutions?

Which parts of the proposed solution were most effective in reducing the resource usage as compared to the existing solutions?  The batch timeout tuning?  The policy for sorting incoming requests?  Optimization of the Orchestrator component?

**Reviewer Confidence:**

3: The reviewer is confident but not certain that the evaluation is correct

**Scope:**

4: The work is relevant to the Web and to the track, and is of broad interest to the community

---

### Official Review · Reviewer_2ofE · 2023-11-23

**Novelty:** 6
**Technical Quality:** 6

**Review:**

This is an interesting paper on a relevant topic. The text is clear. The experimental evaluation is well thought out and thorough.

There is (I think) a minor slip in the explanation around (2). This is very minor but the symbol h_v^{l+1} is used for the representation of vertex v in layer l+1 both before and after update making an equation that is not an equation. Either use two symbols or the := instead of =. (This is a really minor point.)

While I'm very familiar with graph computation and streaming graph systems I'm not familiar with the specifics of GNN. There are some things in the paper that are hard for me to understand.

Equation 4 is written in a slightly confusing way. An assumption is that U_{r_i} is none of the r_1,... r_j in (3) -- which makes sense in context.  Again this is a minor point.

(12) should be Constraints I presume. There is also a misspelling "indecates" nearby.

I like the experimental set up. It's good to test three workloads and three GNN back ends. The timeout configuration made a lot of difference to the lambdagrapher performance.

**Questions:**

In section 2.1 the GNN inference is described using N(v) as the neighbour set of v and there is no dependence on layer. However section 2.2 states there are layer level fluctuations "represented by the difference in the number of vertices at each GNN layer". How are these reconciled? Is this simply that some vertices no longer need to perform the computation?

Presumably in real situations ties in (4) are common (if the ratio is 1 because all the nodes in the request are present in several buffers). How are ties resolved?
In figure 12 we see CPU usage approximately double with tweaks to timeout. I think that the 11c was performed using adaptive timeout. It seems this parameter can really make a huge difference. To what extent is it possible that similar tweaks to GraphLearn and AWSGNN could halve their CPU usage?

**Reviewer Confidence:**

2: The reviewer is willing to defend the evaluation, but it is likely that the reviewer did not understand parts of the paper

**Scope:**

4: The work is relevant to the Web and to the track, and is of broad interest to the community

---

### Decision · Program_Chairs · 2024-01-22

**Decision:**

Accept (Oral)

**Comment:**

The paper addresses a new serverless system for GNN inference with resource efficiency. The reviewers are mostly in agreement to the merits of this paper -- topical relevance, interesting and non-trivial problem, well-crafted solutions addressing operator sharing and resource allocation, and extensive evaluation. In the meantime, the reviewers do have concerns about ablation studies and feature selectivity, and practical issues such as computation time, maintainability, adoption barriers, evolving data etc.

 Overall, given the ratings and issues, I believe that this paper's merits outweigh its concerns. As the area chair, I'd like to recommend acceptance of this paper.